# Combination therapy for mCRPC with immune checkpoint inhibitors, ADT and vaccine: A mathematical model

**Nourridine Siewe** [ID]¹ᵒ*, Avner Friedman²ᵒ

**1** School of Mathematical Sciences, College of Science, Rochester Institute of Technology, Rochester, New York, United States of America, **2** Mathematical Biosciences Institute & Department of Mathematics, The Ohio State University, Columbus, Ohio, United States of America

ᵒ These authors contributed equally to this work.
* nourridine@aims.ac.za

**Data Availability Statement:** All relevant data are within the manuscript and its Supporting information files.

**Funding:** This research was supported by the Dean's Research Initiative Grant #15874 of the 390

## Abstract

Metastatic castration resistant prostate cancer (mCRPC) is commonly treated by androgen deprivation therapy (ADT) in combination with chemotherapy. Immune therapy by checkpoint inhibitors, has become a powerful new tool in the treatment of melanoma and lung cancer, and it is currently being used in clinical trials in other cancers, including mCRPC. However, so far, clinical trials with PD-1 and CTLA-4 inhibitors have been disappointing. In the present paper we develop a mathematical model to assess the efficacy of any combination of ADT with cancer vaccine, PD-1 inhibitor, and CTLA-4 inhibitor. The model is represented by a system of partial differential equations (PDEs) for cells, cytokines and drugs whose density/concentration evolves in time within the tumor. Efficacy of treatment is determined by the reduction in tumor volume at the endpoint of treatment. In mice experiments with ADT and various combinations of PD-1 and CTLA-4 inhibitors, tumor volume at day 30 was always larger than the initial tumor. Our model, however, shows that we can decrease tumor volume with large enough dose; for example, with 10 fold increase in the dose of anti-PD-1, initial tumor volume will decrease by 60%. Although the treatment with ADT in combination with PD-1 inhibitor or CTLA-4 inhibitor has been disappointing in clinical trials, our simulations suggest that, disregarding negative effects, combinations of ADT with checkpoint inhibitors can be effective in reducing tumor volume if larger doses are used. This points to the need for determining the optimal combination and amounts of dose for individual patients.

## 1 Introduction

Prostate cancer is a major public health concern in the United States, with 248,000 new cases annually, and 34,000 deaths [1]. In metastatic prostate cancer, 5-year survival is 35% [2]. Androgen is a group of sex hormones that give men their 'male' characteristics. A major sex hormone is testosterone which is produced mainly in the testes. Prostate cells need androgen

College of Science, and the SEED Grant #16067, at Rochester Institute of Technology. 391 This work was also supported by the Mathematical Biosciences Institute of The Ohio 392 State University. There was no additional external funding received for this study.

**Competing interests:** The authors have declared that no competing interests exist.

for their growth and function [3, 4]. Androgen affects the immune system by increasing the proliferation of T regulatory cells (Tregs) through secretion of IL-10 [3, 5, 6]. Testoterone, upon entering prostate cells, is enzymatically converted into a more potent androgen, dihydro-testoterone (DHT), which binds to androgen receptor with more affinity [7].

When cancer cells undergo necrosis, they release high mobility group box-1 (HMGB-1) which activates dendritic cells (DCs) [8–10]. Activated DCs mature as antigen presenting cells (APCs) and play a critical role in the communication between the innate and adaptive immune responses. Once activated, DCs produce IL-12, which activates effector T cells $CD4^+$ Th1 and $CD8^+$ T [11, 12]. Th1 produces IL-2 which further promotes proliferation of the effector T cells. Both $CD4^+$ Th1 and $CD8^+$ T cells kill cancer cells [13–15]. $CD8^+$ T cells are more effective in killing cancer cells, but the helper function of $CD4^+$ Th1 cells improves the efficacy of tumor-reactive $CD8^+$ T cells [16].

Cancer vaccines serve to enlarge the pool of tumor-specific T cells from the naive repertoire, and also to activate tumor specific T cells which are dormant [17]. GM-CSF can activate dendritic cells, and is commonly used as a cancer vaccine [18–20].

PD-1 is an immunoinhibitory receptor predominantly expressed on activated T cells [21, 22]. Its ligand PD-L1 is upregulated on the same activated T cells, and in some human cancer cells [21, 23]. The compex PD-1-PD-L1 is known to inhibit T cells function [22]. Immune checkpoints are regulatory pathways in the immune system that inhibit its active response against specific targets. In case of cancer, the complex PD-1-PD-L1 functions as an immune checkpoint for anti-tumor T cells. CTLA-4 is another immunoinhibitory receptor expressed on activated T cells, the complex CTLA-4-B7 acts as a checkpoint inhibitor for anti-tumor T cells [24, 25]. There has been much progress in recent years in developing checkpoint inhibitors, primarily anti-PD-1 and anti-PD-L1 (e.g., Nivolumab) [26], and anti-CTLA-4 (e.g., Ipilimumab) [27, 28].

The standard care of metastatic prostate cancer is androgen deprivation therapy (ADT), commonly referred to as medical castration. Under ADT, blood tests show that patients develop adaptive immunity [29], and the level of effective T cells (Th1 and $CD8^+$ T cells) increases. Enzalutamide (ENZ) is anti-androgen drug (approved in 2018) that inhibits androgen binding to androgen receptor on prostate cells, and it also inhibits androgen receptor from entering into the nucleus [30]. Clinical trials show that ENZ has significantly longer progression-free and overall survival than 'standard care' of androgen suppression [31]. ENZ is administered orally, once daily, with tablets or capsules [32].

In this paper we consider metastatic castration resistant prostate cancer (mCRPC), that is, metastatic prostate cancer with androgen-independent cancer cells. Sipuleucel-T (Provenge) (Sip-T) is a cancer vaccine (approved in 2010) for treatment of men with symptomatic or minimally symptomatic mCRPC. The vaccine is made by drawing immune cells from patients and culturing them with combinant fusion protein containing prostatic acid phosphotase (PAP) and GM-CSF. It is administered intravenously to activate dendritic cells [33], which indirectly increases antigen-specific T cells [34, 35].

Treatments of mCRPC include ADT in combination with chemotherapeutic drugs [36, 37], and current clinical trials include also cancer vaccines and immune therapy, primary checkpoint inhibitors [38–41].

Treatment of mCRPC with ADT and PD-1 inhibitor has been disappointing [42], conferring only modest benefits [43], even though PD-L1 is increased under ADT [44]. Treatment with ADT and CTLA-4 inhibitor was also disappointing, since it did not increase the overall survival time [42]. Challenges and rationales for immune checkpoint inhibitors in the treatment of mCRPC are discussed in [44].

Preclinical trials with androgen ablation (ADT) and cancer vaccine show increase in both CD8+ T cells and Tregs [45]. Such a combination therapy is most effective when vaccine is delivered after ADT [46, 47].

Vaccine Sip-T activates dendritic cells, and hence indirectly activates T cells. When ligand B7 on the activated dendritic cells combines with CTLA-4 or effective T cells, it initiates a signaling cascade that blocks the activation and proliferation of the T cells. This suggests that a combination therapy with ADT, Sip-T and CTLA-4 or PD-1 inhibitors may be effective in treatment of mCRPC.

Ardiani et al. [48] treated prostate cancer in mice with a combination of ENZ and a vaccine that targets the Twist antigen (involved in the epithelial-to-mesenchymal transition and metastasis) and increases the functional Twist-specific CD8+ T cells. ENZ was found to be immune inert since no changes were seen in CD4+ T cell proliferation and Treg functional assays, and ENZ did not also diminish the Twist vaccine's ability to generate CD4+ and CD8+ Twist-specific T cells responses. However, the combination of ENZ with Twist vaccine resulted in significantly increased overall survival of the mice compared to treatments with Twist vaccine alone (27.5 weeks vs 10.3 weeks). This suggests that combination of ENZ and immunotherapy is a promising treatment strategy for mCRPC.

In other mice experiments, Shen et al. [49] found that combination of ADT with anti-PD-1 and/or anti-CTLA-4 significantly delayed the development of castration resistance, reduced tumor volume and prolonged survival of tumor-bearing mice in some cases. Immunotherapy alone did not improve survival, and was ineffective if not administered in the peri-castration period.

There have been several clinical trials examining the effect of checkpoint inhibitors in combination with ADT and Sip-T for the treatment of mCRPC. A list of clinical trials that are currently in progress in phases I–III is given in de Almeida et al. [39]; they include anti-PD-1 (NCT03506997), anti-CTLA-4 (NCT01498978), anti-PD-1+anti-CTLA-4 (NCT02601014), anti-PD-1+Sip-T (NCT03024216), anti-CTLA-4+Sip-T (NCT01804465), anti-PD-1+ENZ (NCT04116775), anti-CTLA-4+ENZ (NCT01688492), and anti-PD-1+anti-CTLA-4+vaccine (Sip-T) (NCT02616185). Monotherapy with anti-CTLA-4 or anti-PD-1 in clinical trials did not improve tumor growth in most cases. Mathematical models of prostate cancer that consider treatment with androgen deprivation are reviewed in a number of papers (e.g., [50–52]); models with intermittent androgen ablation strategies aimed to reduce androgen resistance were developed in [50, 53], where additional references are given.

There are several mathematical models of combination therapy with checkpoint inhibitors, for either generic or specific cancers. Lai and Friedman [54] considered combination therapy for melanoma with BRAF and PD-1 inhibitors. They showed that the combination is effective, in terms of tumor volume reduction, in "small" doses, but not in "large" doses. In [55] they considered treatment of a generic tumor with cancer vaccine (GVAX) and anti-PD-1. The vaccine produces GM-CSF which promotes activation of anti-cancer T cells. They addressed the question of which dose amounts and proportions to inject in order to increase synergy and efficacy. In another paper [56] they considered combination of PD-1 inhibitor with oncolytic virus (OV); the virus infects only cancer cells and replicates in them. Since CD8+ T cells kill both infected and uninfected cancer cells, they may either promote or suppress the tumor. They showed that anti-PD-1 in dose $\gamma_P$ in combination with OV in dose $\gamma_O$ is anti-cancer for one set of pairs $(\gamma_P, \gamma_O)$, while in the complementary set the combination is pro-cancer.

In [57] they considered combination of PD-1 and VEGF inhibitors and addressed the question in which order to administer the drugs in cases where VEGF inhibitor is known to affect the perfusion of other drugs. They showed that non-overlapping schedule of injections of the

two drugs is significantly more effective than simultaneous injections. In Lai et al. [57] they considered treatment of breast cancer with CTLA-4 inhibitor in combination with BET inhibitor. They noted that more effective combinations to reduce the tumor volume result in higher level of toxicity, as measured by overexpression of TNF-$\alpha$.

Cancer resistance was considered in Lai et al. [58] and Siewe and Friedman [59]. In [58] it was shown that anti-TNF-$\alpha$ reduces cancer resistance to anti-PD-1, and it is more effective if injected after anti-PD-1 injection, rather than simultaneously. In [59] it was shown that initial resistance to anti-PD-1, which is quite common, can be overcome by combination with TGF-$\beta$ inhibitor, but the efficacy of the combination depends on two specific biomarkers.

ENZ inhibits androgen ($A$). It also inhibits androgen receptor (AR) from entering into the nucleus, which we take, in the model, as inhibiting AR. For simplicity, we shall simplify these two different activities of ENZ by combining "androgen" with "androgen receptor", and referring to it as androgen/receptor (A/AR) or, briefly, as androgen $A$.

In the present paper we develop a mathematical model to explore the efficacy of different combination therapies. The model includes androgen-dependent prostate cancer cells ($N$) and androgen-independent (castration-resistant) cancer cells ($M$), dendritic cells ($D$), Th1 cells ($T_1$), CD8$^+$ T cells ($T_8$), T regulatory cells (Tregs, or $T_r$), and cytokines IL-12 ($I_{12}$), IL-10 ($I_{10}$) and IL-2 ($I_2$); the model includes also checkpoints PD-1 and CTLA-4 and their ligands PD-L1 and B7, respectively, and drugs. The $M$ cells are cancer cells that underwent changes (e.g., epigenetic) so that they are adapted to survive and proliferate with (or little) androgen; for simplicity we refer to them as mutated cancer cells.

Androgen blockade increases the death rate of $N$ cells [60] and the mutation rate of $N$ to $M$ [61–63]. Dendritic cells ($D$) are activated by the high mobility group box 1 (HMGB-1) expressed on necrotic cancer cells [9, 10, 64]. The activated dendritic cells secrete pro-inflammatory cytokine $I_{12}$ which induces the differentiation of naive T cells into $T_1$ cells and $T_8$ cells [11, 12, 65, 66], a process inhibited by $I_{10}$ [67] and $T_r$ cells [68]. $I_{10}$ is secreted by cancer cells [3, 5, 6] and by Tregs [69, 70], and Tregs differentiate from naive T cells under activation by Fox3p$^+$ transcription factor, a process enhanced by $I_{10}$ [69, 70]. $T_1$ cells secrete cytokine IL-2 ($I_2$) which enhances the proliferation of $T_1$ and $T_8$ cells.

PD-1 and PD-L1 are expressed on T cells, and PD-L1 is expressed also on cancer cells. The complex PD1/PD-L1 blocks the anti-cancer activity of $T_1$ and $T_8$ cells [71], but also increases the proliferation of $T_r$ by mediating a phenotype change from $T_1$ to $T_r$ [72, 73]. CTLA-4 is expressed on T cells, and B7 is expressed dendritic cells. The complex CTLA-4/B7 blocks the anti-cancer activity of $T_1$ and $T_8$ cells [74], and at the same time it also increases the proliferation of $T_r$ [75]; we assume that this increase in $T_r$ is caused by a change from $T_1$ to $T_r$ phenotype, as in the case of PD-1/PD-L1.

In this paper, we develop for the first time a mathematical model for cancer therapy that combines checkpoint inhibitors, vaccine and chemical castration. The mathematical model is represented by a system of partial differential equations based on Fig 1, which is a network describing the interactions among the cells, cytokines and checkpoints. The list of variables used in the model is given in Table 1. The list includes the following drugs: anti-PD-1, $A_1$ (nivolumab); anti-CTLA-4, $A_4$ (ipilimumab); ENZ ($E$) and Sip-T ($S$).

## 2 Mathematical model

The mathematical model is based on the network shown in Fig 1, with variables listed in Table 1. The variables satisfy a system of partial differential equations in a domain $\Omega(t)$, the region occupied by the cancer cells, which varies with time $t$.

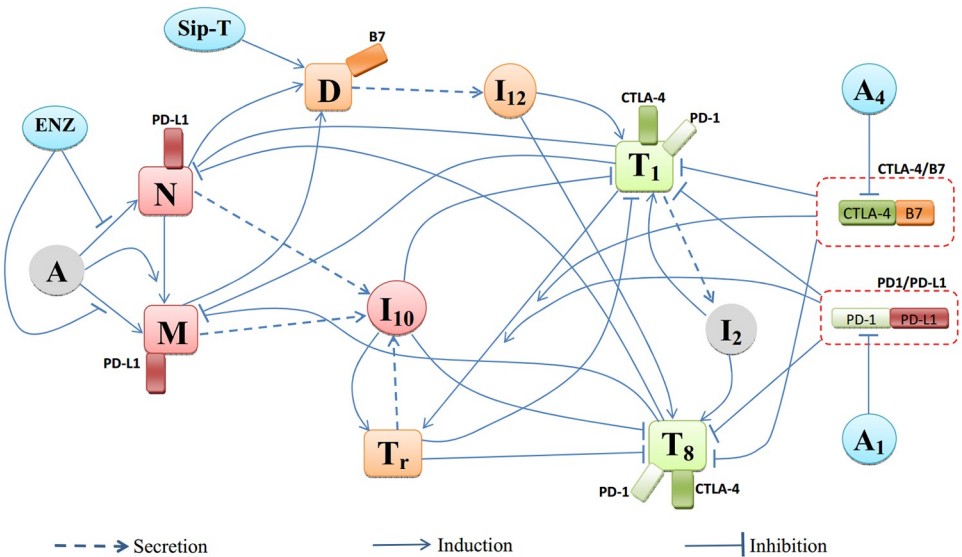

**Fig 1. Network describing the interactions among cells and cytokines under treatment with anti-PD-1, anti-CTLA-4, ENZ and Sip-T.**

We assume that the combined densities of cells within the prostate tumor $\Omega(t)$ remains constant in space and time:

$$N + M + D + T_1 + T_8 + T_r = \theta, \tag{1}$$

for some constant $\theta > 0$. We assume that the densities of immature dendritic cells and naive CD4$^+$ and CD8$^+$ T cells remain constant throughout the tumor tissue. Under Assumption (1), proliferation of cancer cells and immigration of immune cells into the tumor, give rise to internal pressure which results in cells movement with velocity, **u**; **u** depends on space and time and will be taken in units of cm/day. We assume that cytokines and anti-tumor drugs are diffusing within the tumor, and that also cells undergo diffusion (i.e., dispersion).

In what follows, we denote by $Y\frac{X}{K_X+X}$ a quantity proportional to the rate of proliferation/activation of species $Y$ by species X, and by $Y\frac{1}{1+X/K_X}$ the rate proportional to the inhibition of $Y$ by X. If $Y$ is activated by two species, $X_1$ and $X_2$, then we separately add each of the

**Table 1. Variables of the model.** All concentrations are in units of g/cm$^3$.

| Variables | Descriptions | Variables | Descriptions |
|---|---|---|---|
| $D$ | density of dedritic cells | $T_1$ | density of Th1 cells |
| $T_8$ | density of CD8$^+$ T cells | $T_r$ | density of Treg cells |
| $N$ | density of androgen-dependent cancer cells | $M$ | density of mutated (androgen-independent) cancer cells |
| $A$ | concentration of androgen | $I_2$ | concentration of IL-2 |
| $I_{10}$ | concentration of IL-10 | $I_{12}$ | concentration of IL-12 |
| $B_7$ | concentration of B7 | $P_A$ | concentration of CTLA-4 |
| $P_D$ | concentration of PD-1 | $P_L$ | concentration of PD-L1 |
| $Q_1$ | concentration of PD-1/PD-L1 | $Q_2$ | concentration of CTLA-4/B7 |
| $A_1$ | concentration of anti-PD-1 | $A_4$ | concentration of anti-CTLA-4 |
| $E$ | concentration of ENZ | $S$ | concentration of Sip-T |

activated terms, but if $Y$ is inhibited by $X_1$ and $X_2$, then its total inhibition is proportional to $Y \frac{1}{1+X_1/\hat{K}_{X_1}} \frac{1}{1+X_2/\hat{K}_{X_2}}$.

## Equation for androgen-dependent cancer cells ($N$)

We assume a logistic growth for the androgen-dependent cancer cells with carrying capacity $K_{NM}$, to account for the competition for space and nutrients among cancer cells. Androgens are primary regulators of prostate cancer cell growth and proliferation [60]. We accordingly model cancer cell growth rate $\beta$ as an increasing saturating function of $A$, taking

$$\beta(A) = \frac{A}{A_0 + A},$$

where $A_0$ is a level that corresponds to physiologically normal androgen concentration [76].

The drug ENZ ($E$) inhibits androgen binding to androgen receptor [30]. We represent its effect by multiplying $\beta(A)$ by a factor $\frac{1}{1 + E/\hat{K}_E}$, where $\hat{K}_E$ is constant. Androgen-dependent cancer cells $N$ mutate into androgen-independent cells $M$, at a rate that increases with decreasing androgen level [61–63]; we take this mutation rate to be proportional to $\frac{1}{1 + A/\hat{K}_A}$.

Additional mutation from $N$ to $M$ results from the blockade of androgen receptor by ENZ, which we take to be proportional to $\left(1 - \frac{1}{1 + E/\hat{K}_E}\right)$.

Cancer cells are killed by CD8$^+$ T cells [77, 78]. We write the equation for $N$ in the following form:

$$\frac{\partial N}{\partial t} + \nabla \cdot (\mathbf{u}N) - \delta_N \nabla^2 N = \underbrace{\lambda_N \frac{\beta(A)}{1 + E/\hat{K}_E} N \left(1 - \frac{N+M}{K_{NM}}\right)}_{\text{Growth of } N} - \underbrace{\frac{\lambda_{NM}}{1 + A/\hat{K}_A} N}_{N \to M \text{ by low } A}$$

$$\underbrace{- \lambda_N \beta(A) \frac{E/\hat{K}_E}{1 + E/\hat{K}_E N} \left(1 - \frac{N+M}{K_{NM}}\right)}_{N \to M \text{ due to } A-\text{blockade by ENZ}} - \underbrace{\mu_{T_8 N} T_8 N}_{\text{killing by } T_8} - \underbrace{\mu_N \frac{K_A + A}{A} N}_{\text{death}} \quad (2)$$

where $\delta_N$ is the diffusion coefficient, $\mu_{T_8 N}$ is the killing rate of cancer cells by $T_8$ and $\mu_N$ is the natural death rate of cancer cells.

## Equation for mutated androgen-independent tumor cells ($M$)

The dynamics of mutated androgen-independent cancer cells is given by

$$\frac{\partial M}{\partial t} + \nabla \cdot (\mathbf{u}M) - \delta_M \nabla^2 M = q \left( \underbrace{\frac{\lambda_{NM}}{1 + A/\hat{K}_A} N}_{N \to M \text{ by low } A} + \underbrace{\lambda_N \beta(A) \frac{E/\hat{K}_E}{1 + E/\hat{K}_E} N \left(1 - \frac{N+M}{K_{NM}}\right)}_{N \to M \text{ due to } A-\text{blockade by ENZ}} \right)$$

$$- \underbrace{\mu_{T_8 M} T_8 M}_{\text{killing by } T_8} - \underbrace{\mu_M M}_{\text{death}} \quad (3)$$

where $\delta_M$ is a diffusion coefficient, $q$ is their growth rate as they are recruited from the mutation of $N$, $\mu_{T_8 M}$ is the killing rate of cancer cells by $T_8$, and their death rate is independent of

androgen. Note that independent proliferation of castration-resistant cancer cells are included in the term $q\lambda_{NM}/(1 + A/\hat{K}_A)$ when $A$ is small.

## Equation for dendritic cells ($D$)

The binding of extracellular high mobility box 1 (HMGB-1) to toll-like receptor 4 (TLR4) convert the immature dendritic cells, $D_0$, into the activated tumor-associated dendritic cells [9, 10, 64] at a rate proportional to HMGB-1/($H_0$+HMGB-1), where $H_0$ is constant. Assuming that the concentration of HMGB-1 is proportional to the density of cancer cells, this activation rate is proportional to a linear combination of $\dfrac{N}{K_N + N}$ and $\dfrac{M}{K_M + M}$, where $K_N$ and $K_M$ are constants. The vaccine Sip-T ($S$) augments the activation of dendritic cells [33] by a factor $\lambda_{DS} S/(K_S + S)$, for some constants $\lambda_{DS}$, $K_S$. The dynamics of dendritic cells is given by

$$\frac{\partial D}{\partial t} + \nabla \cdot (\mathbf{u}D) - \delta_D \nabla^2 D = D_0 \left( \underbrace{\lambda_{DN} \frac{N}{K_N + N} + \lambda_{DM} \frac{M}{K_M + M}}_{\text{activation by HMGB-1}} + \underbrace{\lambda_{DS} \frac{S}{K_S + S}}_{\text{activation by Sip-T}} \right) - \underbrace{\mu_D D}_{\text{death}}, \quad (4)$$

where $\delta_D$ is a diffusion coefficient, $\mu_D$ is the death rate of dendritic cells, and the activation rates $\lambda_{DN}$ and $\lambda_{DC}$ are constants.

## Equation for Th1 cells ($T_1$)

Naive CD4$^+$ T cells, $T_{10}$, differentiate into Th1 cells under IL-12 inducement [11, 12, 65], and this process is inhibited by IL-10 [67] and Tregs [68]. The proliferation of activated CD4$^+$ T cells is enhanced by IL-2 [79]. Activation and proliferation of $T_1$ cells are inhibited by the complex PD-1/PD-L1 ($Q_1$), represented by a factor $\frac{1}{1+Q_1/\hat{K}_{TQ_1}}$ [71], and by the complex CTLA-4/B7 ($Q_2$) as a factor $\frac{1}{1+Q_2/\hat{K}_{TQ_2}}$ [74]. The complex $Q_1$ also mediates phenotype change from $T_1$ cells to Tregs [72, 73], at a rate $\lambda_{T_r Q_1} T_1 \frac{Q_1}{K_{Q_1} + Q_1}$, and $Q_2$ enhances naive Th cells to become Tregs [75], at a rate $\lambda_{T_r Q_2} T_1 \frac{Q_2}{K_{Q_2} + Q_2}$. Hence $T_1$ satisfies the following equation:

$$\frac{\partial T_1}{\partial t} + \underbrace{\nabla \cdot (\mathbf{u}T_1)}_{\text{advection}} - \underbrace{\delta_T \nabla^2 T_1}_{\text{diffusion}} =$$

$$\left( \lambda_{T_1 I_{12}} T_{10} \underbrace{\frac{I_{12}}{K_{I_{12}} + I_{12}}}_{\text{activation by IL-12}} \cdot \underbrace{\frac{1}{1 + I_{10}/\hat{K}_{I_{10}}}}_{\text{inhibition by IL-10}} \cdot \underbrace{\frac{1}{1 + T_r/\hat{K}_{T_r}}}_{\text{inhibition by Tregs}} + \underbrace{\lambda_{T_1 I_2} T_1 \frac{I_2}{K_{I_2} + I_2}}_{\text{IL-2-induced proliferation}} \right) \quad (5)$$

$$\times \underbrace{\frac{1}{1 + Q_1/\hat{K}_{TQ_1}}}_{\text{inhibition by } Q_1} \cdot \underbrace{\frac{1}{1 + Q_2/\hat{K}_{TQ_2}}}_{\text{inhibition by } Q_2} - \underbrace{T_1 \left( \lambda_{T_r Q_1} \frac{Q_1}{K_{Q_1} + Q_1} + \lambda_{T_r Q_2} \frac{Q_2}{K_{Q_2} + Q_2} \right)}_{(Q_1, Q_2)\text{-induced } T_1 \to T_r \text{ transition}} - \underbrace{\mu_{T_1} T_1}_{\text{death}}.$$

## Equation for CD8$^+$ T cells ($T_8$)

Inactive CD8$^+$ T cells, $T_{80}$, are activated by IL-12 [11, 12, 66], and this process is resisted by IL-10 [67] and Tregs [68]. IL-2 enhances the proliferation of activated CD8$^+$ T cells [79]. Both processes of activation and proliferation are inhibited by PD-1/PD-L1, by a factor $\frac{1}{1+Q_1/\hat{K}_{TQ_1}}$,

and by CTLA-4/B7, by a factor $\frac{1}{1+Q_2/\hat{K}_{TQ_2}}$. Hence, $T_8$ satisfies the following equation:

$$\frac{\partial T_8}{\partial t} + \nabla \cdot (\mathbf{u}T_8) - \delta_T \nabla^2 T_8 =$$

$$\left( \underbrace{\lambda_{T_8 I_{12}} T_{80} \frac{I_{12}}{K_{I_{12}} + I_{12}}}_{\text{activation by IL-12}} \cdot \underbrace{\frac{1}{1 + I_{10}/\hat{K}_{I_{10}}}}_{\text{inhibition by IL-10}} \cdot \underbrace{\frac{1}{1 + T_r/\hat{K}_{T_r}}}_{\text{inhibition by Tregs}} + \underbrace{\lambda_{T_8 I_2} T_8 \frac{I_2}{K_{I_2} + I_2}}_{\text{IL-2-induced proliferation}} \right)$$

$$\times \underbrace{\frac{1}{1 + Q_1/\hat{K}_{TQ_1}}}_{\text{inhibition by }Q_1} \cdot \underbrace{\frac{1}{1 + Q_2/\hat{K}_{TQ_2}}}_{\text{inhibition by }Q_2} - \underbrace{\mu_{T_8} T_8}_{\text{death}}. \tag{6}$$

## Equation for Tregs ($T_r$)

Naive CD4$^+$ T cells differentiate into Tregs under activation by Fox3p+ transcription factor, a process enhanced by IL-10 [69, 70]. We have the following equation for $T_r$:

$$\frac{\partial T_r}{\partial t} + \nabla \cdot (\mathbf{u}T_r) - \delta_T \nabla^2 T_r =$$

$$\underbrace{\lambda_{T_r I_{10}} T_{10} \frac{I_{10}}{K_{I_{10}} + I_{10}}}_{I_{10}\text{-enhanced naive T cells activation}} + \underbrace{T_1 \left( \lambda_{T_r Q_1} \frac{Q_1}{K_{Q_1} + Q_1} + \lambda_{T_r Q_2} \frac{Q_2}{K_{Q_2} + Q_2} \right)}_{(Q_1, Q_2)\text{-induced } T_1 \rightarrow T_r \text{ transition}} - \underbrace{\mu_{T_r} T_r}_{\text{death}}, \tag{7}$$

where the second term in the right-hand side is the same as in Eq (5).

## Equation for IL-2 ($I_2$)

Cytokine IL-2 is produced by activated Th1 cells [79]. Hence,

$$\frac{\partial I_2}{\partial t} - \delta_{I_2} \nabla^2 I_2 = \underbrace{\lambda_{I_2 T_1} T_1}_{\text{secretion by CD4}^+ \text{ T cells}} - \underbrace{\mu_{I_2} I_2}_{\text{degradation}}. \tag{8}$$

## Equation for IL-10 ($I_{10}$)

Cytokine IL-10 is produced by cancer cells [3, 5, 6] and Tregs [69, 70]. Hence IL-10 satisfies the following equation:

$$\frac{\partial I_{10}}{\partial t} - \delta_{I_{10}} \nabla^2 I_{10} = \underbrace{\lambda_{I_{10}N} N + \lambda_{I_{10}M} M + \lambda_{I_{10}T_r} T_r}_{\text{secretion by } N, M \text{ and } T_r} - \underbrace{\mu_{I_{10}} I_{10}}_{\text{degradation}}. \tag{9}$$

## Equation for IL-12 ($I_{12}$)

The pro-inflammatory cytokine IL-12 is secreted by activated dendritic cells [11, 12], so that

$$\frac{\partial I_{12}}{\partial t} - \delta_{I_{12}} \nabla^2 I_{12} = \underbrace{\lambda_{I_{12}D} D}_{\text{secretion by DCs}} - \underbrace{\mu_{I_{12}} I_{12}}_{\text{degradation}}. \tag{10}$$

### Equations for androgen ($A$)

Androgen is consumed by prostate cancer cells $N$ at a rate proportional to $\beta(A)E$ [3, 80]. Hence, $A$ satisfies the following equation

$$\frac{\partial A}{\partial t} - \delta_A \nabla^2 A = \underbrace{\lambda_A}_{\text{production of } A} - \underbrace{\mu_{NA} N \beta(A) E}_{\text{consumption by} N} - \underbrace{\mu_A A}_{\text{decay}} \tag{11}$$

where $\lambda_A$ is the constant production rate and $\mu_A$ is the degradation rate.

### Equations for PD-1 ($P_D$), PD-L1 ($P_L$) and PD-1/PD-L1 ($Q_1$)

PD-1 is expressed on the membrane of activated CD4$^+$ T cells, activated CD8$^+$ T cells. We assume that the number of PD-1 proteins per cell is the same for $T_1$, $T_r$ and $T_8$ cells. If we denote by $\rho_{P_D}$ the ratio between the mass of the PD-1 proteins in one T cell to the mass of the cell, so that

$$P_D = \rho_{P_D}(T_1 + T_8 + T_r).$$

The coefficient $\rho_{P_D}$ is constant when no anti-PD-1 drug is administered. In this case, to a change in $T = T_1 + T_8 + T_r$, given by $\partial T/\partial t$, there corresponds a change in $P_D$, given by $\rho_{P_D}\partial T/\partial t$. For the same reason, $\nabla \cdot (\mathbf{u}P_D) = \rho_{P_D}\nabla \cdot (\mathbf{u}T)$ and $\nabla^2 P_D = \rho_{P_D}\nabla^2 T$ when no anti-PD-1 drug is injected. Hence, $P_D$ satisfies the equation:

$$\frac{\partial P_D}{\partial t} + \nabla \cdot (\mathbf{u}P_D) - \delta_T \nabla^2 P_D = \frac{\partial(T_1 + T_8 + T_r)}{\partial t} + \nabla \cdot (\mathbf{u}(T_1 + T_8 + T_r)) - \delta_T \nabla^2(T_1 + T_8 + T_r).$$

Recalling Eqs (5)–(7) for $T_1$, $T_8$ and $T_r$, we get

$$\begin{aligned}
\frac{\partial P_D}{\partial t} + \quad & \nabla \cdot (\mathbf{u}P_D) - \delta_T \nabla^2 P_D = \\
& \rho_{P_D} \Bigg\{ \left[ (\lambda_{T_1 I_{12}} T_{10} + \lambda_{T_8 I_{12}} T_{80}) \frac{I_{12}}{K_{I_{12}} + I_{12}} \cdot \frac{1}{1 + I_{10}/\hat{K}_{I_{10}}} \cdot \frac{1}{1 + T_r/\hat{K}_{T_r}} \cdot \right. \\
& + (\lambda_{T_1 I_2} T_1 + \lambda_{T_8 I_2} T_8) \frac{I_2}{K_{I_2} + I_2} \Bigg] \frac{1}{1 + Q_1/\hat{K}_{TQ_1}} \cdot \frac{1}{1 + Q_2/\hat{K}_{TQ_2}} \\
& - T_1 \left( \lambda_{T_r Q_1} \frac{Q_1}{K_{Q_1} + Q_1} + \lambda_{T_r Q_2} \frac{Q_2}{K_{Q_2} + Q_2} \right) \\
& + \left( \lambda_{T_r I_{10}} T_{10} \frac{I_{10}}{K_{I_{10}} + I_{10}} + T_1 \left( \lambda_{T_r Q_1} \frac{Q_1}{K_{Q_1} + Q_1} + \lambda_{T_r Q_2} \frac{Q_2}{K_{Q_2} + Q_2} \right) \right) \\
& . - (\mu_{T_1} T_1 + \mu_{T_8} T_8 + \mu_{T_r} T_r) \Bigg\}.
\end{aligned}$$

When anti-PD-1 drug ($A_1$) is applied, PD-1 is depleted at a rate proportional to $A_1$, and, in this case, the ratio $P_D/(T_1 + T_8 + T_r)$ may change. In order to include in the model both cases of with and without anti-PD-1, we replace $\rho_{P_D}$ in the above equation by $P_D/(T_1 + T_8 + T_r)$.

Hence,

$$
\frac{\partial P_D}{\partial t} + \nabla \cdot (\mathbf{u} P_D) - \delta_T \nabla^2 P_D =
$$

$$
\frac{P_D}{T_1 + T_8 + T_r} \Bigg\{ \Bigg[ (\lambda_{T_1 I_{12}} T_{10} + \lambda_{T_8 I_{12}} T_{80}) \frac{I_{12}}{K_{I_{12}} + I_{12}} \cdot \frac{1}{1 + I_{10}/\hat{K}_{I_{10}}} \cdot \frac{1}{1 + T_r/\hat{K}_{T_r}} \cdot
$$

$$
+ (\lambda_{T_1 I_2} T_1 + \lambda_{T_8 I_2} T_8) \frac{I_2}{K_{I_2} + I_2} \Bigg] \frac{1}{1 + Q_1/\hat{K}_{TQ_1}} \cdot \frac{1}{1 + Q_2/\hat{K}_{TQ_2}}
$$

$$
- T_1 \left( \lambda_{T_r Q_1} \frac{Q_1}{K_{Q_1} + Q_1} + \lambda_{T_r Q_2} \frac{Q_2}{K_{Q_2} + Q_2} \right)
$$

$$
+ \left( \lambda_{T_r I_{10}} T_{10} \frac{I_{10}}{K_{I_{10}} + I_{10}} + T_1 \left( \lambda_{T_r Q_1} \frac{Q_1}{K_{Q_1} + Q_1} + \lambda_{T_r Q_2} \frac{Q_2}{K_{Q_2} + Q_2} \right) \right)
$$

$$
\cdot - (\mu_{T_1} T_1 + \mu_{T_8} T_8 + \mu_{T_r} T_r) \Bigg\} - \underbrace{\mu_{P_D A_1} P_D A_1}_{\text{depletion by anti}-PD-1}, \tag{12}
$$

where $\mu_{P_D A_1}$ is the depletion rate of PD-1 by anti-PD-1.

We assume that the number of PD-L1 proteins in one $T_1$ cell is the same as in one $T_r$ cell and one $T_8$ cell, and denote by $\rho_{P_L}$ the ratio of the mass of all the PD-L1 proteins in one $T_1$ cell to the mass of one cell. We assume that this ratio on cancer cells is $\rho_{P_L} \varepsilon_C$. Hence,

$$
P_L = \rho_{P_L} [T_1 + T_8 + T_r + \varepsilon_C (N + M)]. \tag{13}
$$

PD-L1 from T cells or cancer cells combines with PD-1 on the plasma membrane of T cells, forming a complex PD-1/PD-L1 ($Q_1$) on the T cells [21, 23]. Denoting the association and disassociation rates of $Q_1$ by $\alpha_{P_D P_L}$ and $\mu_{Q_1}$, respectively, we write

$$
P_D + P_L \underset{\mu_{Q_1}}{\overset{\alpha_{P_D P_L}}{\rightleftharpoons}} Q_1.
$$

Since the half-life of $Q_1$ is less than 1 second (i.e., $1.16 \times 10^{-5}$ day) [81], we may approximate the dynamical equation for $Q_1$ by the steady state equation $\alpha_{P_D P_L} P_D P_L = \mu_{Q_1} Q_1$, or

$$
Q_1 = \sigma_1 P_D P_L, \tag{14}
$$

where $\sigma_1 = \alpha_{P_D P_L}/\mu_{Q_1}$.

**Equation for CTLA-4 ($P_A$), B7 ($B_7$) and CTLA-4/B7 ($Q_2$).** CTLA-4 is a receptor expressed on activated $T_1$ and $T_8$ cells [82] and the complex CTLA-4/B7 blocks the activities of these cells [74, 82]. CTLA-4 is constitutively expressed on $T_r$ cells, but its activity is not blocked by the complex CTLA-4/B7 [83]. We assume that the number of CTLA-4 proteins per cell is the same for $T_1$ and $T_8$ cells, but different for $T_r$ cells, by a factor $\kappa_T$. We denote by $\rho_{P_A}$ the ratio between the mass of all CTLA-4 proteins in one T cell to the mass of this cell, so that

$$
P_A = \rho_{P_A} (T_1 + T_8 + \kappa_T T_r).
$$

The coefficient $\rho_{P_A}$ is constant when no anti-CTLA-4 drug is administered. In this case, to a change in $T = T_1 + T_8 + T_r$, given by $\partial T/\partial t$, there corresponds a change of $P_A$, given by

$\rho_{P_A} \partial T / \partial t$. Similar changes in $P_A$ arises from the terms of diffusion and advection, so that

$$
\begin{aligned}
\frac{\partial P_A}{\partial t} + & \nabla \cdot (\mathbf{u} P_A) - \delta_T \nabla^2 P_A = \\
& \rho_{P_A} \Bigg\{ \left[ (\lambda_{T_1 I_{12}} T_{10} + \lambda_{T_8 I_{12}} T_{80}) \frac{I_{12}}{K_{I_{12}} + I_{12}} \cdot \frac{1}{1 + I_{10}/\hat{K}_{I_{10}}} \cdot \frac{1}{1 + T_r/\hat{K}_{T_r}} \cdot \right. \\
& \left. + (\lambda_{T_1 I_2} T_1 + \lambda_{T_8 I_2} T_8) \frac{I_2}{K_{I_2} + I_2} \right] \frac{1}{1 + Q_1/\hat{K}_{TQ_1}} \cdot \frac{1}{1 + Q_2/\hat{K}_{TQ_2}} \\
& - T_1 \left( \lambda_{T_r Q_1} \frac{Q_1}{K_{Q_1} + Q_1} + \lambda_{T_r Q_2} \frac{Q_2}{K_{Q_2} + Q_2} \right) \\
& + \kappa_T \left( \lambda_{T_r I_{10}} T_{10} \frac{I_{10}}{K_{I_{10}} + I_{10}} + T_1 \left( \lambda_{T_r Q_1} \frac{Q_1}{K_{Q_1} + Q_1} + \lambda_{T_r Q_2} \frac{Q_2}{K_{Q_2} + Q_2} \right) \right) \\
& . - (\mu_{T_1} T_1 + \mu_{T_8} T_8 + \kappa_T \mu_{T_r} T_r) \Bigg\}.
\end{aligned}
$$

When anti-CTLA-4 drug ($A_4$) is applied, CTLA-4 is depleted at a rate proportional to $A_4$, and, in this case, the ratio $P_A/(T_1 + T_8 + \kappa_T T_r)$ may change. In order to include in the model both cases, with and without anti-CTLA-4, we replace $\rho_{P_A}$ in the above equation by $P_A/(T_1 + T_8 + \kappa_T T_r)$. Hence,

$$
\begin{aligned}
\frac{\partial P_A}{\partial t} + & \nabla \cdot (\mathbf{u} P_A) - \delta_T \nabla^2 P_A = \\
& \frac{P_A}{T_1 + T_8 + \kappa_T T_r} \Bigg\{ \left[ (\lambda_{T_1 I_{12}} T_{10} + \lambda_{T_8 I_{12}} T_{80}) \frac{I_{12}}{K_{I_{12}} + I_{12}} \cdot \frac{1}{1 + I_{10}/\hat{K}_{I_{10}}} \cdot \frac{1}{1 + T_r/\hat{K}_{T_r}} \cdot \right. \\
& \left. + (\lambda_{T_1 I_2} T_1 + \lambda_{T_8 I_2} T_8) \frac{I_2}{K_{I_2} + I_2} \right] \frac{1}{1 + Q_1/\hat{K}_{TQ_1}} \cdot \frac{1}{1 + Q_2/\hat{K}_{TQ_2}} \\
& - T_1 \left( \lambda_{T_r Q_1} \frac{Q_1}{K_{Q_1} + Q_1} + \lambda_{T_r Q_2} \frac{Q_2}{K_{Q_2} + Q_2} \right) \\
& + \kappa_T \left( \lambda_{T_r I_{10}} T_{10} \frac{I_{10}}{K_{I_{10}} + I_{10}} + T_1 \left( \lambda_{T_r Q_1} \frac{Q_1}{K_{Q_1} + Q_1} + \lambda_{T_r Q_2} \frac{Q_2}{K_{Q_2} + Q_2} \right) \right) \\
& . - (\mu_{T_1} T_1 + \mu_{T_8} T_8 + \kappa_T \mu_{T_r} T_r) \Bigg\} - \mu_{P_A A_4} P_A A_4,
\end{aligned}
\tag{15}
$$

where $\mu_{P_A A_4}$ is the depletion rate of CTLA-4 by anti-CTLA-4.

The ligand B7 is expressed on dendritic cells, so that

$$
B_7 = \rho_{B_7} D, \quad \rho_{B_7} = \text{constant}.
$$

CTLA-4 and B7 from the complex CTLA-4/B7 ($Q_2$) with association and disassociation rates $\alpha_{P_A B_7}$ and $\mu_{Q_2}$, respectively:

$$P_A + B_7 \underset{\mu_{Q_2}}{\overset{\alpha_{P_A B_7}}{\rightleftharpoons}} Q_2.$$

We assume that the half-life of $Q_2$ is very short [81, 84], so that we may approximate the dynamics $Q_2$ by the steady state, $\alpha_{P_A B_7} P_A B_7 = \mu_{Q_2} Q_2$, or

$$Q_2 = \sigma_2 P_A B_7,$$

where $\sigma_2 = \alpha_{P_A B_7}/\mu_{Q_2}$.

**Equations for anti-PD-1 ($A_1$) and anti-CTLA-4 ($A_4$).** If a drug $X$ with dose $\gamma_X$ and half-life $t_{1/2}$ is injected at time $t_0$, we assume that its effect at time $t$ ($t > t_0$) continues to be effective at level $\gamma_X e^{-\alpha t}$, where $e^{-\alpha t_{1/2}} = 1/2$, i.e., $\alpha = \dfrac{\ln 2}{t_{1/2}}$.

We shall compare our simulations with experimental results in [49], where PD-1 inhibitor and CTLA-4 inhibitor were injected at fixed dose in days 0, 3 and 6. The half-life of PD-1 inhibitor (nivolumab) is 26.7 days [85], and $A_1$ is depleted in the process of blocking PD-1, hence

$$\frac{\partial A_1}{\partial t} - \delta_{A_1} \nabla^2 A_1 = \underbrace{\gamma_{A_1} f_{A_1}(t)}_{\text{source}} - \underbrace{\mu_{P_D A_1} P_D A_1}_{\text{depletion through blocking PD}-1} - \underbrace{\mu_{A_1} A_1}_{\text{degradation}} \tag{16}$$

where

$$f_{A_1}(t) = \begin{cases} e^{-\frac{\ln 2}{26.7}t}, & \text{for } 0 \le t < 3, \\ e^{-\frac{\ln 2}{26.7}t} + e^{-\frac{\ln 2}{26.7}(t-3)}, & \text{for } 3 \le t < 6, \\ e^{-\frac{\ln 2}{26.7}t} + e^{-\frac{\ln 2}{26.7}(t-3)} + e^{-\frac{\ln 2}{26.7}(t-6)}, & \text{for } 6 \le t \le 30. \end{cases}$$

The half-life of CTLA-4 (ipilimumab) is 14.7 days [86], hence

$$\frac{\partial A_4}{\partial t} - \delta_{A_4} \nabla^2 A_4 = \underbrace{\gamma_{A_4} f_{A_4}(t)}_{\text{source}} - \underbrace{\mu_{P_A A_4} P_A A_4}_{\text{depletion through blocking CTLA}-4} - \underbrace{\mu_{A_4} A_4}_{\text{degradation}} \tag{17}$$

where

$$f_{A_4}(t) = \begin{cases} e^{-\frac{\ln 2}{14.7}t}, & \text{for } 0 \le t < 3, \\ e^{-\frac{\ln 2}{14.7}t} + e^{-\frac{\ln 2}{14.7}(t-3)}, & \text{for } 3 \le t < 6, \\ e^{-\frac{\ln 2}{14.7}t} + e^{-\frac{\ln 2}{14.7}(t-3)} + e^{-\frac{\ln 2}{14.7}(t-6)}, & \text{for } 6 \le t \le 30. \end{cases}$$

**Equation for ENZ ($E$).** In [49] the ADT drug was degarelix ($G$) and it was injected once every 30 days. The half-life of degarelix is 53 days [87], so its effective level at time $t$ is $\gamma_G e^{-\frac{\ln 2}{53}t}$, where $\gamma_G$ is the initial dose, with average $0.7\gamma_G$. In our model we let ENZ ($E$) take the role of degarelix. The drug ENZ has similar effect as degarelix, but is somewhat different in its mechanisms, and its half-life is 5.8 days [32]. In mice experiment [48] it was given in a way that maintained the level of daily dose ($\gamma_E$) constant. Since $E$ is depleted in the process of inhibiting

androgen, we have:

$$\frac{\partial E}{\partial t} - \delta_E \nabla^2 E = \underbrace{\gamma_E}_{\text{source}} - \underbrace{\mu_{AE} A E}_{\text{depletion through blocking androgen}} - \underbrace{\mu_E E}_{\text{degradation}} \tag{18}$$

**Equation for Sip-T ($S$).** In mCRPC clinical trials [88] Sip-T was administered with three infusions, two weeks apart. We approximate the effective level of the dose by a constant $\gamma_S$. The drug is depleted in the process of activating dendritic cells, so that

$$\frac{\partial S}{\partial t} - \delta_S \nabla^2 S = \underbrace{\gamma_S}_{\text{source}} - \underbrace{\mu_{DS} D_0 S}_{\text{depletion through activating dendritic cells}} - \underbrace{\mu_S S}_{\text{degradation}} \tag{19}$$

## Equation for cells velocity (u)

We assume that all cells have approximately the same diffusion coefficient. Adding Eqs (2)–(7) and using Eq (1), we get

$$\theta \times \nabla \cdot \mathbf{u} = \sum_{j=2}^{7} [\text{RHS of Eq. (2.j)}]. \tag{20}$$

To simplify the computations, we assume that the tumor is spherical, and that all the densities and concentrations are radially symmetric, that is, functions of $(r, t)$, $0 \leq r \leq R(t)$, where $r = R(t)$ is the boundary of the tumor, and that $\mathbf{u} = u(r, t)\mathbf{e}_r$, where $\mathbf{e}_r$ is the unit radial vector.

**Equation for the free boundary ($R$).** We assume that the free boundary $r = R(t)$ moves with the velocity of cells, so that

$$\frac{dR(t)}{dt} = u(R(t), t). \tag{21}$$

**Boundary conditions.** We assume that the inactive CD4$^+$ and CD8$^+$ T cells that migrated from the lymph nodes into the tumor microenvironment have constant densities $\hat{T}_1$ and $\hat{T}_8$, respectively, at the tumor boundary, and that they are activated by IL-12 upon entering the tumor. We then have the following conditions at the tumor boundary:

$$\begin{aligned} \frac{\partial T_1}{\partial t} &+ \sigma_0 \frac{I_{12}}{K_{I_{12}} + I_{12}} (T_1 - \hat{T}_1)^+ = 0, \\ \frac{\partial T_8}{\partial r} &+ \sigma_0 \frac{I_{12}}{K_{I_{12}} + I_{12}} (T_8 - \hat{T}_8)^+ = 0 \quad \text{at } r = R(t). \end{aligned} \tag{22}$$

We impose no-flux boundary condition on all the remaining variables:

No flux for $N$, $M$, $D$, $T_r$, $I_2$, $I_{10}$, $I_{12}$, $P_A$, $P_D$, $A_1$, $A_4$, $E$ and $S$ at $r = R(t)$; (23)

it is tacitly assumed here that the receptors PD-1 and CTLA-4, and ligands PD-L1 and B7 become active only after the T cells are already inside the tumor.

## 3 Numerical simulations

All the computations were done using Python 3.5.4. The parameter values of the model equations are estimated in S1 File Section 1 and are listed in S1 File Tables 1 and 2. Parameter

sensitivity analysis was performed in S1 File Section 2, and the techniques used for the simulations are in described in S1 File Section 3.

### 3.1 Model calibration

We simulated the model Eqs (2)–(21) with boundary conditions (23) and initial conditions, in units of g/cm$^3$,

$$N = 0.16, \ M = 8 \times 10^{-3}, \ D = 3.5 \times 10^{-3}, \ T_1 = 8 \times 10^{-5}, \ T_8 = 1.6 \times 10^{-4}, \ T_r = 4 \times 10^{-5},$$

$$I_2 = 1.2 \times 10^{-12}, \ I_{10} = 3 \times 10^{-10}, \ I_{12} = 7 \times 10^{-10}, \ A = 2.2 \times 10^{-10}, \ R = 0.5 \text{ cm}.$$

We let the program run for 5 days ($t = -5$ to $t = 0$) before we began therapy. Fig 2 shows the profiles of the average densities/concentrations of the variables of the model, and of the tumor volume, with/without ADT. Without ENZ, the density of mutated cells (M) remains small, and tumor volume grows exponentially. With ENZ, given daily from $t = 0$ to $t = 30$, the tumor volume is first increasing, then decreasing during days 2–21, and finally it is again increasing. These changes in monotonicity can be explained by the fact that there is sharp decrease in androgen-dependent density ($N$) and slow increase in androgen-independent density ($M$) during an intermediate period, as seen in Fig 2.

Fig 3 displays the densities of T cells, DCs, PD-1 and CTLA-4, at 3 time points represented by the dots in Fig 2 and identified by 'Pre-C', 'ENZ Effective' and 'C-Resistant'.

In Fig 4A, we simulated the profile of tumor volume under treatment with various combinations of anti-PD-1, anti-CTLA-4 and Sip-T, and in Fig 4B, we added ENZ, with the same protocol as in Figs 2 and 3. We see that adding one or two drugs in any of the combinations increases the efficacy.

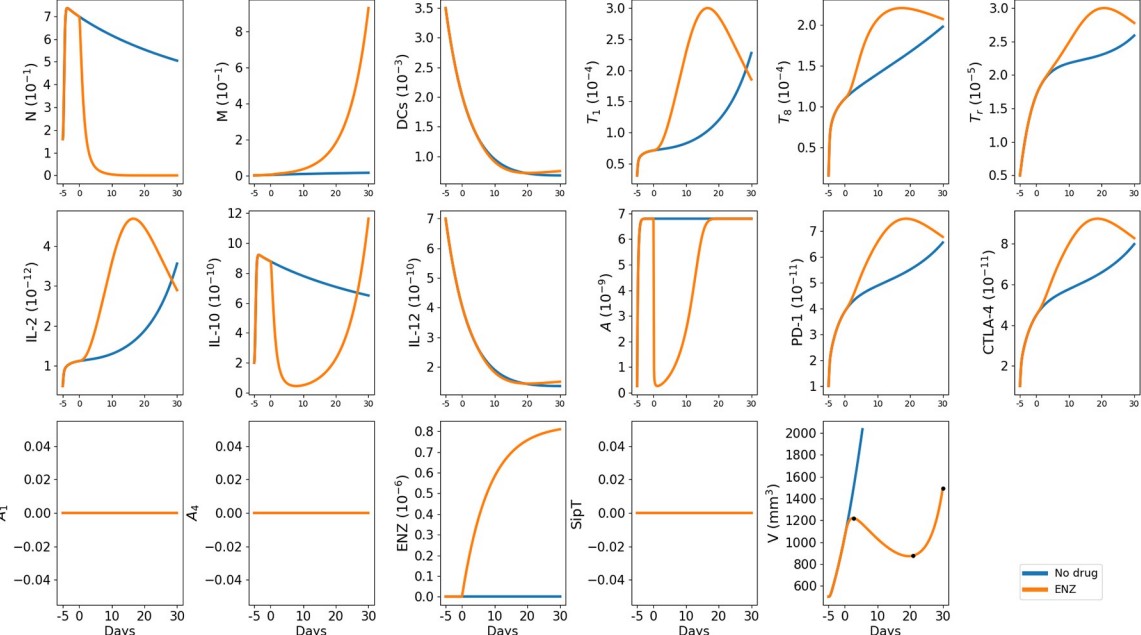

**Fig 2. Simulation of the average densities/concentrations of the variables for model (2)–(21) with/without ENZ (ADT) at $\gamma_E = 10^{-7}$ g/cm$^3$·d.** The dots in the 'V' panel represent species' tracking time points as shown in Fig 3. All parameters are as in S1 File Tables 1 and 2. The units of the variables are g/cm$^3$.

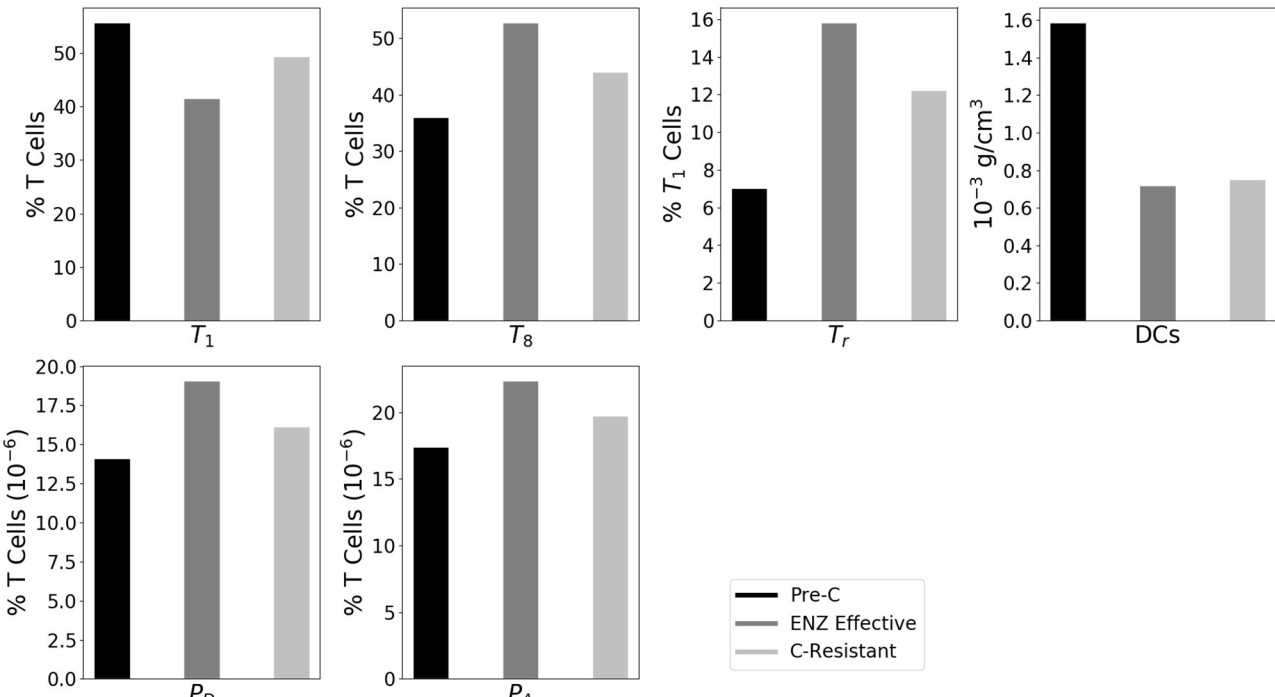

**Fig 3. Cellular immune components of the pre-castration and post-castration within the tumor.** All parameters are as in S1 File Tables 1 and 2. "Pre-C" represents the level of the species at the time when the tumor volume attains its first maximum before decline due to ENZ, with $\gamma_E = 10^{-7}$ g/cm$^3$·d, "ENZ-Effective" is the level of the species at the time when the tumor volume attains its lowest value under ENZ, and "C-Resistant" represents the level of the species at day 30 of treatment with ENZ, when androgen-resistance cells density ($M$) is at highest level.

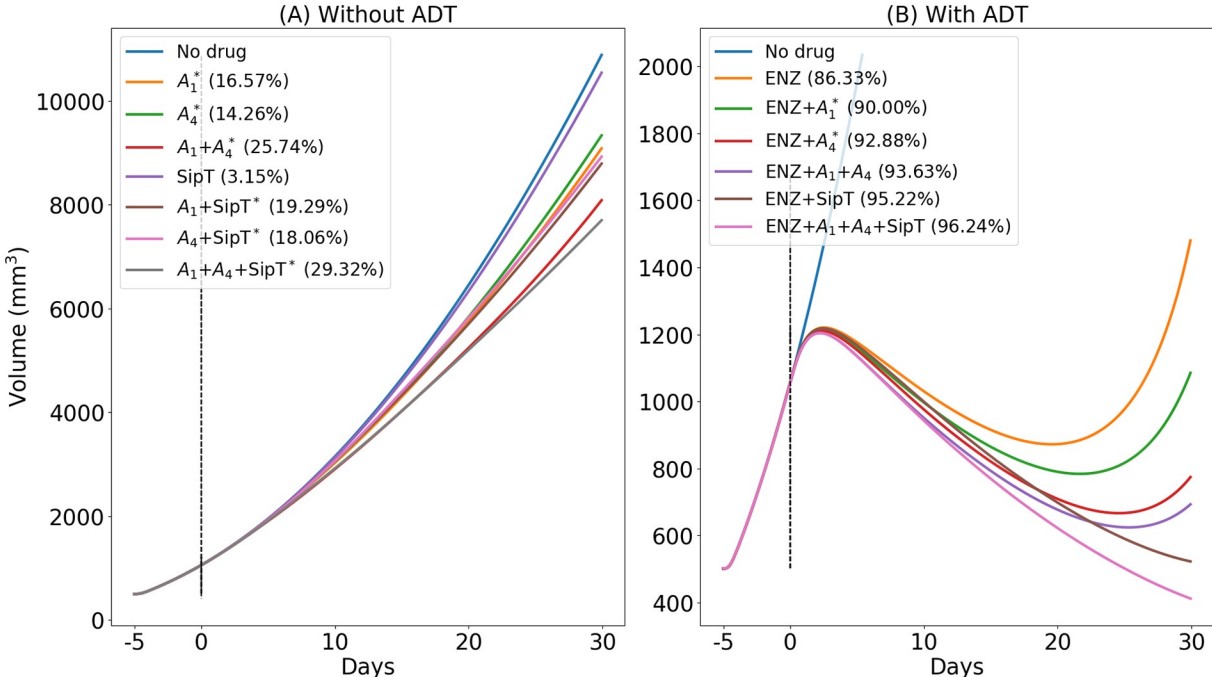

**Fig 4. Simulation of the average densities/concentrations of the variables for model (2)–(21) with ENZ, Sip-T, anti-PD-1 ($A_1$) and anti-CTLA-4 ($A_4$).** The "%" represents the percentage decrease relative to no-treatment at day 30; the symbol "*" indicates treatments which are currently undergoing clinical trials. All parameters are as in S1 File. Tables 1 and 2, with $\gamma_{A_1} = 4 \times 10^{-9}$, $\gamma_{A_4} = 2 \times 10^{-8}$, $\gamma_E = 10^{-7}$ and $\gamma_S = 10^{-6}$, in g/cm$^3$·d.

Degarelix is an androgen-receptor antagonist, which can be viewed as somewhat similar to ENZ in our model. Shen et al. [49] conducted mice experiments with treatment of prostate cancer using degarelix. The levels of T cells and DCs in Fig 3 are in qualitative agreement with Fig 3A in [49], and the levels of PD-1 and CTLA-4 are in qualitative agreement with Fig 4B in [49]. More precisely: As in [49], the level of DCs is decreasing through days 0 (Pre-C), 7 (ENZ Effective), 30 (C-Resistant); $T_1$ is decreasing-increasing; $T_r$, $P_D$ and $P_A$ are increasing-decreasing. The profile of $T_8$ is increasing-decreasing while in [49] the profile of $T_8$ is constant; however, in [49] they also include the profile of NK cells which is increasing-decreasing while in our model we did not include NK, and, instead, let $T_8$ be the only cells who kill cancer cells. Hence, the $T_8$ in our model functions as $T_8$ + NK in the experimental results of [49]; and since in [49] NK is increasing-decreasing while $T_8$ is flat, there is a fit of our profile of $T_8$ with [49].

On the other hand, the concentrations of cytokine IL-2 in the microenvironment (outside the tumor) in Fig 5B of [49], cannot be compared with the concentrations in Fig 2, which is taken within the tumor, because of the large diffusion of cytokines.

In Fig 4A, we see that various combinations without ENZ do not reduce tumor volume significantly. This is in agreement with clinical trials referenced in [49]. In Fig 4B, we see that the combinations with ENZ increase efficacy, from 89.08% to 96.52%; the largest benefits are with combination of all the four drugs, $A^1$+$A^4$+ENZ+SipT. In particular, the combination with $A^1$+$A^4$ increases efficacy from 89.08% to 94.41%; this moderate increase is in agreement with Fig 5A of [49], where degarelix was combined with $\alpha$-PD-1 and $\alpha$-CLTA-4 (ND).

We also note that the increase-decrease-increase profiles of the tumor volumes in Figs 2 and 4B are similar to the increase-decrease-increase profiles of tumor volumes in Fig 5A of [49].

## 3.2 Therapy predictions

The parameter $q$ is the ratio of growth rate of $M$ to growth rate of $N$. According to [50], $q$ is slightly smaller than 1 if the concentrations of DHT-activated androgen receptors and of testosterone-activated receptors are both the same for $N$ and $M$. In our model we view $q$ as a "personalized" parameter (a parameter in personalized, or precision, medicine), and let it vary in the interval $0.6 < q < 1.2$.

We consider the case where the ENZ level is constant for 30 days, and it is either delivered as single agent or in combination with $A_1$, $A_4$ or Sip-T by the same protocol as in Fig 4. Fig 5A shows the profile of tumor volume as function of $q$ and time, $0 < t < 30$.

We introduce two definitions to measure the benefit of treatment. Defining $V_{\text{drug}}(t)$ and $V(t)$ as the tumor volume at time $t$ under treatment and without treatment, respectively, the first definition, in terms of efficacy, is the following:

$$\text{Efficacy} = \frac{V(30) - V_{\text{drug}}(30)}{V(30)} \times 100\%. \tag{24}$$

The second definition is in terms of tumor volume reduction (TVR):

$$\text{TVR} = \frac{V(0) - V_{\text{drug}}(30)}{V(0)} \times 100\%. \tag{25}$$

Efficacy tells us how much we can reduce the tumor volume by treatment compared to no treatment; increased efficacy means improved treatment. But even very high efficacy does not inform whether the initial tumor was actually decreased. To get this information we look at TVR. With TVR, the larger it is the more the tumor volume was reduced compared to the

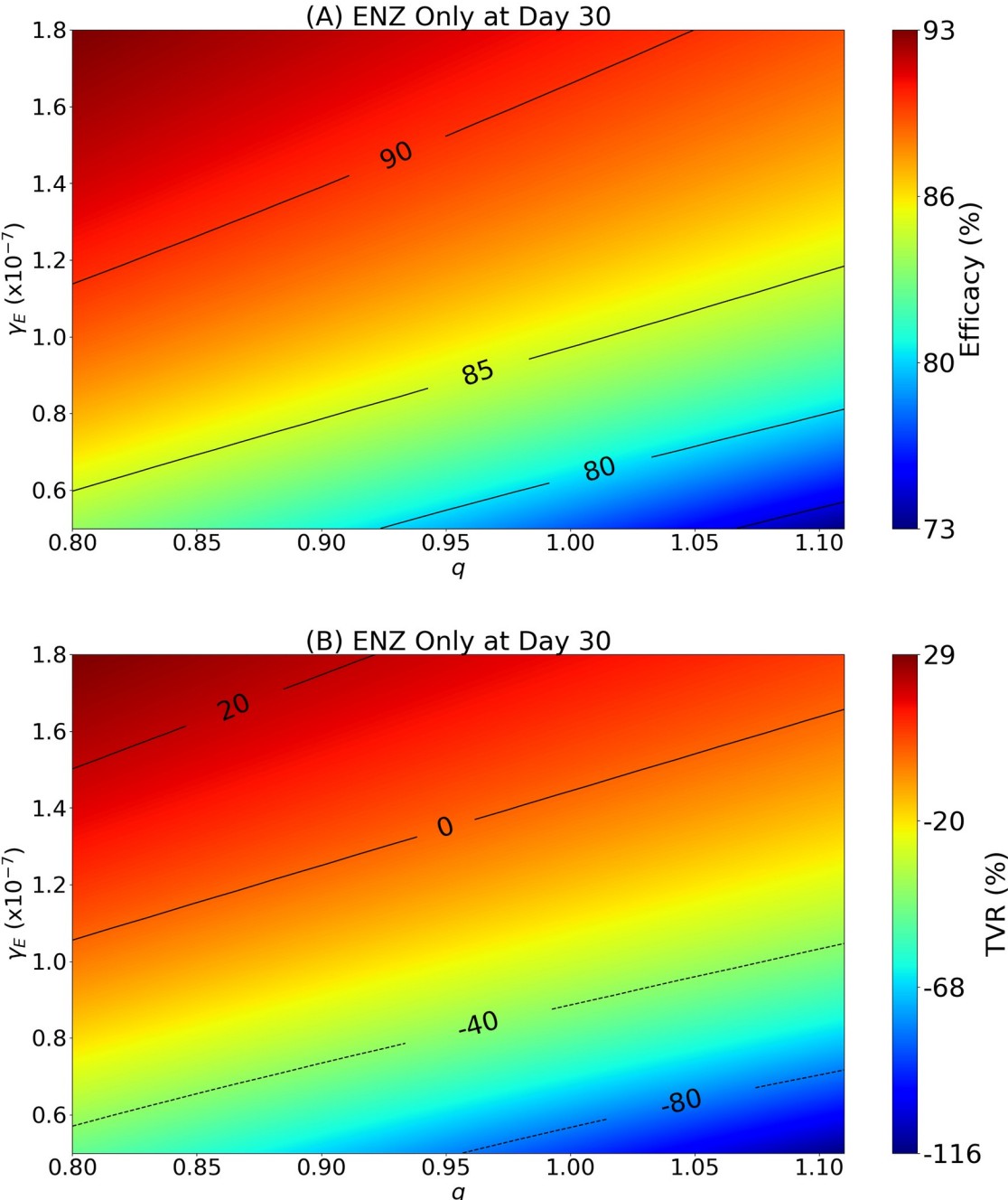

**Fig 5. Benefit maps for treatment with ADT.** To each value of the personalized parameter $q$ and ENZ dose amount $\gamma_E$, the color column in (A) indicates the efficacy, and in (B) indicates the TVR: $0.8 < q < 1.11$ and $\gamma_E$ varies in the range $0.5–1.8\times10^{-7}$ g/cm$^3$·d. In Fig 4: $q = 0.8$, $\gamma_E = 10^{-7}$ g/cm$^3$·d.

initial volume, and TVR negative means that the treatment did not decrease the initial tumor volume. Clearly, a drug that increases efficacy also increases TVR.

Fig 5A is a map showing the benefit of treatment with ENZ, as $\gamma_E$ varies in the range 0.5–$1.8 \times 10^{-7}$ g/cm$^3$·d, and $q$ varies in the range 0.6–1.2. Fig 5B shows a similar map of benefits in terms of TVR. We see that, as $\gamma_E$ is increased and $q$ is decreased, both efficacy and TVR increase. The range in benefits for efficacy is 70–94%, while for TVR it is −138% to 41%; for

$q = 0.8$ (as in Figs 2–4), initial tumor volume will be reduced by approximately 40% (after 30 days) by treatment with $\gamma_E = 1.8 \times 10^{-7}$ g/cm³·d.

Fig 6A is a map of benefits of treatment with combination of ENZ with $A_1$, when $q = 0.8$ (as in Fig 4), $\gamma_{A_1}$ varies from 0 to $40 \times 10^{-9}$ g/cm³·d (which is 10 times the dose amount

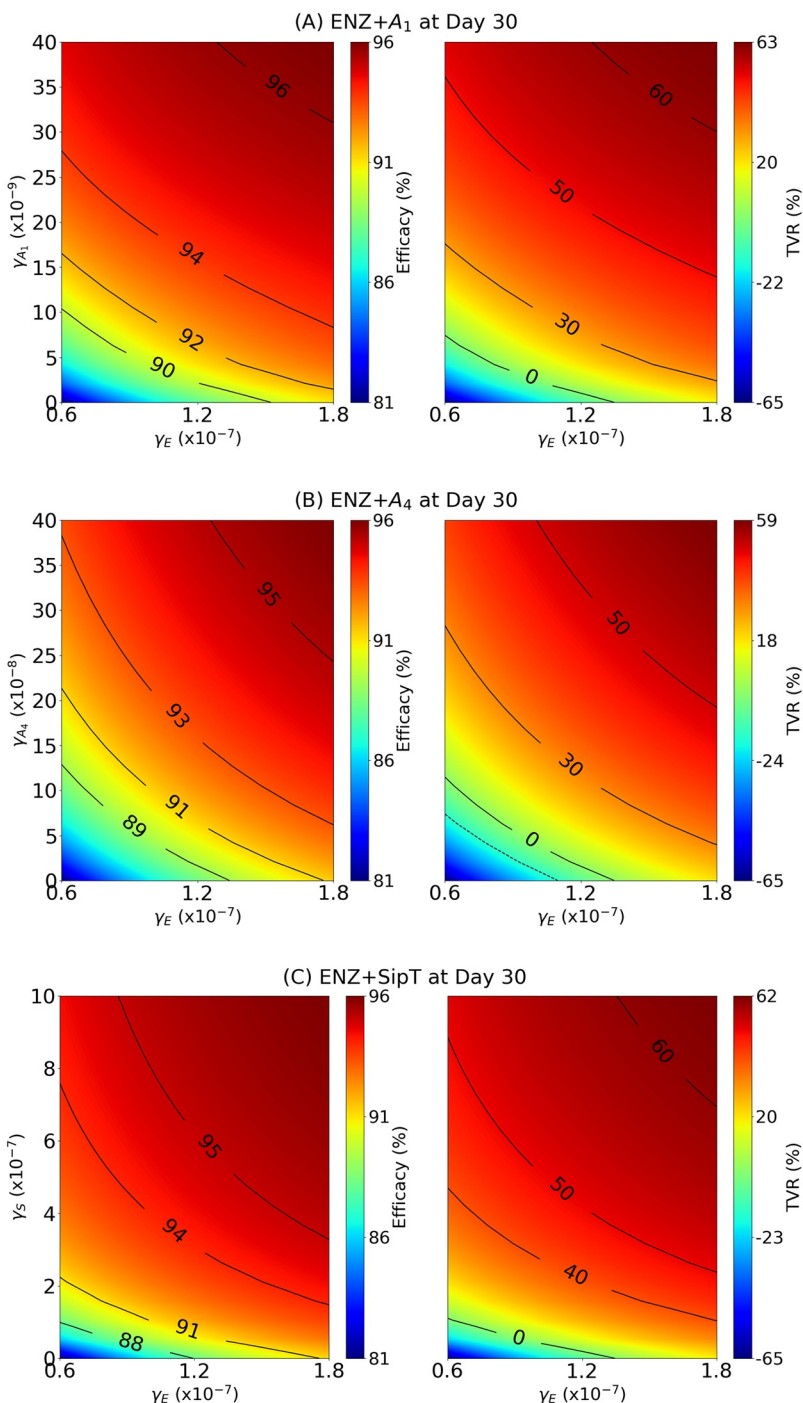

**Fig 6. Benefit maps of combination therapy with ADT.** $\gamma_E$ is in the range 0.6–1.8 × 10⁻⁷ g/cm³·d. (A) $\gamma_E + \gamma_{A_1}$ where $\gamma_{A_1}$ is between 0–40×10⁻⁹ g/cm³·d; (B) $\gamma_E + \gamma_{A_4}$ where $\gamma_{A_4}$ is between 0–40 × 10⁻⁸ g/cm³·d; (C) $\gamma_E + \gamma_S$ where $\gamma_S$ is between 0–10 × 10⁻⁷ g/cm³·d. The color columns indicate the efficacy (on left maps) and TVR (on right maps).

in Fig 4), and $\gamma_E$ varies in the range $0.6 - 1.8 \times 10^{-7}$ g/cm$^3$·d; the dose amount in Fig 4 was $10^{-7}$ g/cm$^3$·d.

We see that efficacy of 95% corresponds, approximately, to 50% of TVR. Keeping $\gamma_E$ at the level of $10^{-7}$ g/cm$^3$·d, as in Fig 4, we can reduce tumor volume by nearly 60% if we increase $\gamma_{A_1}$ by 10 fold of its amount in Fig 4.

The situation in Fig 6B with ENZ+$A_4$ is similar. We can decrease tumor volume by 50% if we increase $\gamma_{A_4}$ 15 fold of its value of $2 \times 10^{-8}$ g/cm$^3$·d in Fig 4.

Fig 6C shows that we can achieve 50% tumor volume reduction with ENZ+$\gamma_S$ if we use half the dose amount that was taken in Fig 4.

## 4 Conclusion

Androgen deprivation therapy (ADT) in combination with chemotherapy significantly increased overall survival time in patients with metastatic prostate cancer [89]. More recently, immune therapy by checkpoint inhibitors, has become a powerful new tool in the treatment of melanoma and lung cancer, and is currently used in clinical trials in other cancers, including metastatic castration resistant prostate cancer (mCRPC). Clinical trials, in increasing number, consider ADT in combination with cancer vaccine and immune checkpoint inhibitors (ICI), particularly for checkpoints CTLA-4 and PD-1 [39]. In the present paper, we developed a mathematical model to assess the efficacy of such combinations, as we vary the dose amounts and proportions of each agent in a combination. The model includes CD4$^+$ and CD8$^+$ T cells, dendritic cells, and cytokines by which these cells interact, as well as cancer cells (androgen-independent ($M$) and androgen-dependent ($N$)), and drugs. The densities/concentrations of these species are evolving within the tumor, and their evolution is described by a system of partial differential equations (PDEs); the tumor region is also evolving in time, and its volume growth is used to assess the effectiveness of treatments.

In previous work on metastatic castration resistant prostate cancer (mCRPC), Jain et al. [50] introduced several parameters as personalized parameters. In the present paper, we introduce one such parameter, $q$, which is the ratio of the growth rate of $M$ cells to the growth rate of $N$ cells.

Simulations of the model for 30 days are shown to be in qualitative agreement with experimental results for mice [49], where we used the same protocol of treatment, and took doses $\gamma_E$ = $10^{-7}$ of ENZ (for ADT), $\gamma_{A_1} = 4 \times 10^{-9}$ (for anti-PD-1), $\gamma_{A_4} = 2 \times 10^{-8}$ (for anti-CTLA-4) in units of g/cm$^3$·d, and $q$ = 0.8. We then proceeded to evaluate (in Fig 6) the effectiveness of various combinations of $\gamma_E$ with $\gamma_{A_1}$, $\gamma_{A_4}$ and $\gamma_S$ (vaccine).

The experimental results in [49] show a tumor volume reduction of only 5–10%. On the other hand, the simulations in Fig 6 show that, in the mice model protocol of [49], we can achieve a much better tumor reduction by increasing the values of $\gamma_E$, $\gamma_{A_1}$, and $\gamma_{A_4}$. In particular, with fixed $\gamma_E$ and $q$ as above, if $\gamma_{A_1}$ is increased 10 fold, the treatment with $(\gamma_E, \gamma_{A_1})$ reduces tumor volume by nearly 60% (at day 30). Similarly, if $\gamma_{A_4}$ is increased 15 fold, the treatment with $(\gamma_E, \gamma_{A_4})$ reduces tumor volume by 50%.

The model has several limitations:

1. We made a simplification by combining androgen with androgen receptor into one variable, which we just referred to it as androgen. This however does not affect the interactions associated with ADT by ENZ.

2. The assumption (1) is another simplification, since it implies that non-cancerous prostate cells within the tumor have constant density, as if they were in homeostasis.

We did not discuss the question whether the PDE system of the model has a solution. This is indeed the case, and be proved by the same method as in [90].

Clinical trials of ADT and immune checkpoint inhibitors have been disappointing [42–44]. The simulations in Fig 6, based on mice experiments, suggest that combination of ADT with PD-1 and CTLA-4 inhibitors would have much more benefits if we increase significantly the dose of these checkpoint inhibitors.

We note however that in terms of clinical applications, PD-1 inhibition is associated with adverse events such as thyroid dysfunction and pneumonitis, CTLA-4 inhibition is closely associated with colitis and hypophysitis, and both drugs are associated with rash and hepatitis [91], and ENZ adverse events includes seizure and ischemic heart disease. This raises the question of determining the maximum dosages, in combinations of ICI and ENZ, that will reduce significantly these side effects. Another question that needs to be addressed in clinical setting is drug resistance, which is primary obstacle to successful cancer treatment. These issues are beyond the scope of the present work. However, the present paper can be used as a first step in addressing these clinical issues.

## Supporting information

**S1 File. Parameters estimates, sensitivity analysis, numerical methods and tables of parameters.**
(PDF)

## Acknowledgments

The authors wish to thank Dr. Tin Phan for reviewing the paper and making many useful suggestions. The funders had no role in study design, data collection and analysis, decision to publish, or preparation of the manuscript.

## Author Contributions

**Conceptualization:** Nourridine Siewe, Avner Friedman.

**Data curation:** Nourridine Siewe, Avner Friedman.

**Formal analysis:** Nourridine Siewe, Avner Friedman.

**Funding acquisition:** Nourridine Siewe.

**Investigation:** Nourridine Siewe, Avner Friedman.

**Methodology:** Nourridine Siewe, Avner Friedman.

**Project administration:** Nourridine Siewe, Avner Friedman.

**Resources:** Nourridine Siewe.

**Software:** Nourridine Siewe.

**Validation:** Nourridine Siewe, Avner Friedman.

**Visualization:** Nourridine Siewe, Avner Friedman.

**Writing – original draft:** Nourridine Siewe, Avner Friedman.

**Writing – review & editing:** Nourridine Siewe, Avner Friedman.

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
