## [Decision Letter · Decision Letter 0]

10 Dec 2021

PONE-D-21-29139Combination Therapy for mCRPC with Immune Checkpoint Inhibitors, ADT and Vaccine: A Mathematical ModelPLOS ONE

Dear Dr. Siewe

Thank you for submitting your manuscript to PLOS ONE. After careful consideration, we feel that it has merit but does not fully meet PLOS ONE’s publication criteria as it currently stands. Therefore, we invite you to submit a revised version of the manuscript that addresses the points raised during the review process.

 Please note that both reviewers have raised several concerns that require attention. Kindly revise the manuscript according to the reviewer comments. 

Please submit your revised manuscript by 12 January 2022. If you will need more time than this to complete your revisions, please reply to this message or contact the journal office at plosone@plos.org. Please include the following items when submitting your revised manuscript:A rebuttal letter that responds to each point raised by the academic editor and reviewer(s). You should upload this letter as a separate file labeled 'Response to Reviewers'.A marked-up copy of your manuscript that highlights changes made to the original version. You should upload this as a separate file labeled 'Revised Manuscript with Track Changes'.An unmarked version of your revised paper without tracked changes. You should upload this as a separate file labeled 'Manuscript'.If applicable, we recommend that you deposit your laboratory protocols in protocols.io to enhance the reproducibility of your results. Protocols.io assigns your protocol its own identifier (DOI) so that it can be cited independently in the future. For instructions see: https://journals.plos.org/plosone/s/submission-guidelines#loc-laboratory-protocols. Additionally, PLOS ONE offers an option for publishing peer-reviewed Lab Protocol articles, which describe protocols hosted on protocols.io. Read more information on sharing protocols at https://plos.org/protocols?utm_medium=editorial-email&utm_source=authorletters&utm_campaign=protocols.

We look forward to receiving your revised manuscript.

Kind regards,

Afsheen Raza, PhD

Academic Editor

PLOS ONE

“This research was supported by the Dean’s Research Initiative Grant #15874 of the College of Science, and the SEED Grant # 16067, at Rochester Institute of Technology. This work was also supported by the Mathematical Biosciences Institute of The Ohio State University. There was no additional external funding received for this study.”

Reviewers' comments:

Reviewer's Responses to Questions

**Comments to the Author**

1. Is the manuscript technically sound, and do the data support the conclusions?

Reviewer #1: Yes

Reviewer #2: Yes

2. Has the statistical analysis been performed appropriately and rigorously? 

Reviewer #1: Yes

Reviewer #2: N/A

3. Have the authors made all data underlying the findings in their manuscript fully available?

Reviewer #1: Yes

Reviewer #2: Yes

4. Is the manuscript presented in an intelligible fashion and written in standard English?

Reviewer #1: Yes

Reviewer #2: Yes

5. Review Comments to the Author

Reviewer #1: What are the implications of these results in clinical applications? And how to use it？Maybe that is something which readers may be more concerned about.Please give a more detailed description in the results and discussion part.

Reviewer #2: Review for “Combination Therapy for mCRPC with Immune Checkpoint Inhibitors, ADT and Vaccine: A Mathematical Model” by Nourridine Siewe and Avner Friedman.

The manuscript provides a detailed formulation of a system of partial differential equations for the treatment of metastatic castration resistant prostate cancer (mCRPC) using a combination of androgen deprivation therapy (ADT), vaccine, and checkpoint inhibitors. The framework itself is built based on previous studies by the authors and others. The authors estimate model parameters using information from literature and show that simulation results agree qualitatively with experimental results in mice. In addition, simulations of different dose combinations suggest possible benefits for the combined treatment of ADT, vaccine, and immune therapy at higher dosages, which has not been observed experimentally. Aside from this interesting hypothesis, the modeling framework itself is a great contribution to science, especially the mathematical and biological modeling communities. Overall, I strongly support the acceptance of this manuscript.

With the understanding that every modeling framework is built based on a series of assumptions, there are several key points regarding the assumptions of the model formulation and the simulation results that I would like the authors to address.

1. Regarding the cancer population and androgen:

a. The proliferation for the androgen-independent (castrate resistant) cancer cells is quite strange. Their production is proportional to the mutation due to evolutionary pressure (Equation 2.3), without any independent production. This is a strong assumption that require further justifications.

b. Prostate cancer is known to eventually develop resistance to whatever treatment used on it. In this framework, the cancer cells are assumed to never develop any resistance to immune and vaccine treatment (Equation 2.2 and 2.3). This has clinical implication on the applicability of the model. For example, without resistance, the strongest dose can be used to suppress the tumor indefinitely.

c. The term “mutation” may not be entirely appropriate in the model formulation. During stressful environment (treatment), there are many methods that the cancer cells can use to improve its chance of survival. For example, reversible epigenetic change (“adaptation”) is perhaps more appropriate. This is especially true when the strength of the “adaption” rate is dependent on the environment as is modeled here (Equation 2.2 and 2.3).

d. Since the model focuses on mCRPC and two treatments other than ADT, the above two points suggest that instead of having a population that depends strongly on androgen (Equation 2.2), which would be negligible in a true mCRPC case, a castrate resistant population (Equation 2.3) is better coupled with other populations that are resistant to the vaccine and immune therapies.

2. Regarding treatment dynamics:

a. The equations for ENZ and SIP-T (Equations 2.18 and 2.19) contain depletion terms based on mass action interaction between the drug and the appropriate target. On other hand, the actual effects of these drugs are represented by saturating functions (Equations 2.2 and 2.4). I hope the authors can clarify this inconsistency or the assumption behind the formulation.

b. The combined effects (from different sources such as drug) are sometimes modelled multiplicatively and other times modelled additively. For example, in Equation 2.4, the activation effect of HMGB-1 and Sip-T is modelled additively, while in Equation 2.5, the activations by IL-12 and the inhibition by IL-10 and Tregs are modelled multiplicatively. What are the rationales behind these choices?

3. A strength of the paper is the meticulous estimation of the parameters, which is complemented by a global sensitivity analysis to determine the importance of some parameters. However, the method of parameter estimation in section 5 leaves much room for uncertainty and the sensitivity analysis result in section 6 is not meant to fill this gap. To better support the quantitative result, I suggest the following steps.

a. Establish the important parameters, perhaps by mean of a global sensitivity analysis on all parameters similar to section 6.

b. Carry out the simulation results (for example, the ones in figure 6) with the upper and lower values of a selected few parameters (by their sensitivity). This will give better confidence in the quantitative results of the paper.

c. LHS-PRCC is meant for parameters whose effect on the variable of interest do not change sign (monotonic relationship). Did the authors examine this condition prior to the application of LHS-PRCC?

4. The interpretation and modelling of the variable A:

a. Equation 2.11 uses a constant production for androgen. However, androgen is produced in a negative feedback loop to maintain an equilibrium level of androgen.

b. The authors interpret A as the androgen level; however, this is somewhat problematic for two main reasons. First, the growth of the cancer cells depend on the bound androgen receptors, which is translocated to the nucleus and integrated to chromosomal DNA (these actions are inhibited by Enzalutamide). Secondly, if A is androgen, then the production of A within the spherical tumor is not realistic (since its main production is elsewhere). In this way, it may be more appropriate to consider A as the activated androgen receptor, which would require a modification of the constant production rate by Enzalutamide.

I also have some minor points:

1. In section 5, the estimated values for the volume, size, and weight of adult men should be for men over the age of 65, if possible.

2. On page 3, second to last paragraph, while reference 50 is one of the major development in the modeling of prostate cancer, since its publication, a large number of models have been introduced. There are several recent comprehensive reviews and comparisons of mathematical models for various aspect of prostate cancer.

3. In the conclusion on page 19, the authors mention the possible side effect of PD-1 and CTLA-4 inhibitors; however, ENZ also has severe side effects that is dose-dependent. An optimal dosage study would benefit from considering these side effects, perhaps similar several previous work by one of the authors.

4. On page 16, first line of subsection 3.2, q is defined as a ratio of growth rate of M to growth rate of N, which means it is a unitless constant. But line 3 of table 2 defines as the proliferation rate of M with a unit of /day (also Equation 2.3).

5. It has been shown that the growth rate of mutated prostate cancer cells are often similar to that of the wild type prostate cancer cells. This means even if q<1 (second to last paragraph on page 27) the value for q would be very close to 1.

6. First paragraph on page 6, cancer cells competes for more than just “space.”

7. Since this is a PDE model with a fixed density (Equation 2.1), what are the implications on the healthy prostate cells which share the same space as the cancer cells?

8. Is there existing work that guarantee the well-posedness of such intricate PDE system, or is it an open question?

6. PLOS authors have the option to publish the peer review history of their article (what does this mean?). If published, this will include your full peer review and any attached files.

Reviewer #1: No

Reviewer #2: **Yes: **Tin Phan

---

## [Editor Report · Decision Letter 1]

26 Dec 2021

Combination Therapy for mCRPC with Immune Checkpoint Inhibitors, ADT and Vaccine: A Mathematical Model

PONE-D-21-29139R1

Dear Dr. Siewe,

We’re pleased to inform you that your manuscript has been judged scientifically suitable for publication and will be formally accepted for publication once it meets all outstanding technical requirements.

Kind regards,

Afsheen Raza, PhD

Academic Editor

PLOS ONE
---

## [Editor Report · Acceptance letter]

30 Dec 2021

PONE-D-21-29139R1 

Combination Therapy for mCRPC with Immune Checkpoint Inhibitors, ADT and Vaccine: A Mathematical Model 

Dear Dr. Siewe:

I'm pleased to inform you that your manuscript has been deemed suitable for publication in PLOS ONE. Congratulations! Your manuscript is now with our production department. 

Kind regards, 

on behalf of

Dr. Afsheen Raza 

Academic Editor

PLOS ONE